# Antioxidant Activity of Planar Catechin Conjugated with Trolox

**DOI:** 10.3390/antiox13101165

**Published:** 2024-09-25

**Authors:** Wakana Shimizu, Yoshimi Shoji, Kei Ohkubo, Hiromu Ito, Ikuo Nakanishi, Kiyoshi Fukuhara

**Affiliations:** 1Division of Organic and Medicinal Chemistry, Showa University School of Pharmacy, Shinagawa-ku, Tokyo 142-8555, Japan; gp21-w010@grad.showa-u.ac.jp; 2Quantum RedOx Chemistry Team, Quantum Life Spin Group, Institute for Quantum Life Science (iQLS), National Institutes for Quantum Science and Technology (QST), Chiba-shi, Chiba 263-8555, Japan; shoji.yoshimi@qst.go.jp (Y.S.); ohkubo@irdd.osaka-u.ac.jp (K.O.); ito.hiromu@qst.go.jp (H.I.); 3Institute for Open and Transdisciplinary Research Initiatives (OTRI), Osaka University, Suita, Osaka 565-0871, Japan

**Keywords:** catechin, trolox, tocopherol, antioxidant activity

## Abstract

Planar catechin (PCat), a natural antioxidant with a fixed 3D catechin structure on a plane, exhibits radical-scavenging activity approximately five times stronger than the conventional catechin. We synthesized a compound, PCat-TrOH, by binding Trolox (TrOH), an α-tocopherol analog, to PCat to enhance its antioxidant effect against oxidative stress, such as lipid peroxidation. TrOH shows radical-scavenging activity about 6.5 times greater than PCat, and PCat-TrOH exhibited a similar level of radical-scavenging activity to TrOH. Additionally, PCat-TrOH demonstrated twice the radical-scavenging activity against reactive oxygen species compared to PCat or TrOH. This compound is also expected to exhibit an excellent antioxidant effect against lipid peroxidation caused by radical chain reactions, through interactions with vitamin C, similar to that in the case of α-tocopherol.

## 1. Introduction

Oxidative stress is associated with the onset of and progression of various diseases such as cancer, diabetes, neurodegenerative diseases (Alzheimer’s disease, Parkinson’s, Huntington’s, and amyotrophic lateral sclerosis), arteriosclerosis, and heart disease [1,2,3,4]. The mitochondria, the main source of reactive oxygen species (ROS), contribute to oxidative stress due to electron leakage in the electron transport chain during energy production [5]. Mitochondrial damage exacerbates this process, leading to the production of even more ROS. Excessive ROS production results in cell damage or cell death. Furthermore, ROS generated by inflamed cells or wounds cause oxidative damage to lipids and proteins in the cell membrane, as well as to enzymes and DNA, leading to various diseases [6]. Therefore, consuming substances that can eliminate ROS and reduce oxidative stress are effective in preventing the onset and progression of diseases associated with oxidative stress [7,8]. Various natural substances can eliminate ROS and suppress oxidative stress [9,10]. The consumption of red wine, known for its effectiveness in preventing coronary artery disease, has been shown to be due to the antioxidant effects of polyphenol compounds, such as resveratrol, catechin, epicatechin, and anthocyanin [11,12]. Among these, catechin is found not only in grapes but also green tea and apples. Catechin, with its catechol structure, reduces and eliminates ROS [9]. It also has a chelating effect on metals, helping to eliminate metals involved in ROS generation [13]. In fact, catechin intake has been reported to have a preventative effect against diseases caused by oxidative stress [14].

Given that (+)-catechin has a structure where the chroman skeleton (AC ring) and catechol skeleton (B ring) are perpendicular to each other, we introduced an isopropyl group at the 3-position of the AC ring and the C6’-position of the C ring to synthesize a planar catechin (PCat) with a fixed 3D structure on a plane [15]. PCat exhibits radical-scavenging activity five times stronger than (+)-catechin and has been shown to possess powerful α-glucosidase inhibition, amyloid β aggregation inhibition, antitumor, and antiviral effects [16].

Tocopherol, a representative fat-soluble antioxidant, suppresses the oxidation of polyunsaturated fatty acids, abundant in biological membranes, into lipid peroxides through radical chain reactions [17]. Tocopherol eliminates free radicals generated during the lipid peroxidation process through reducing actions [18]. Although tocopherol itself is oxidized in this process, vitamin C can reduce tocopherol, allowing it to regain its antioxidant effect [19]. The oxidized vitamin C is then regenerated by glutathione, NADH, and other substances, forming an antioxidant network in the body [20]. This defense mechanism effectively counters many oxidative stresses [21,22,23]. We hypothesized that if a single molecule could have multiple structures with radical-scavenging activity, an antioxidant network could form between these structures, exerting an even more powerful antioxidant effect.

In the present research, we aimed to form oxidative networks within a molecule to exert a more potent antioxidant effect. To achieve this, we synthesized a compound, PCat-TrOH, where PCat was conjugated with Trolox (TrOH), a water-soluble analog of α-tocopherol, and demonstrated its radical-scavenging activity (Figure 1).

## 2. Materials and Methods

### 2.1. General Methods

Unless otherwise noted, all commercially available reagents (Wako Chemicals, Tokyo, Japan; Tokyo Chemical Industry, Tokyo, Japan; and Sigma-Aldrich, St. Louis, MO, USA) were used as received without purification. Solvents for chromatography were of industrial grade, and the processes of all reactions were monitored by thin-layer chromatography that was performed on silica gel 60 F254 (0.25 nm, Merck, Darmstadt, Germany). Column chromatography was performed on silica gel 60 (spherical, 63–210 μm, KANTO CHEMICAL Co. Inc., Tokyo Japan). ^1^H and ^13^C NMR spectra were recorded with a JEOL JNMAL-400 (400 MHz) NMR spectrometer using CDCl_3_, DMSO-*d*_6_ as the solvent, and tetramethylsilane as the internal reference. Mass spectra were measured using a JMS-700V (JEOL) mass spectrometer. The purity of all compounds was determined to be >95% using ^1^H and ^13^C NMR spectroscopy.

### 2.2. Synthesis of PCat-TrOH

#### 2.2.1. Ethyl 3-((6a*S*,12a*R*)-2,3,8,10-tetrahydroxy-5-methyl-5,6a,7,12a-tetrahydroisochromeno[4,3-*b*]chromen-5-yl)propanoate (**1**)

Trimethylsilyl trifluoromethanesulfonate (TMSOTf, 5.42 mL, 30.0 mmol) was slowly added to a solution of (+)-catechin (6.0 g, 20.7 mmol) and ethyl 4-oxovalerate (7.14 mL 51.7 mmol) in dry tetrahydrofuran (THF) (200 mL) at −30 °C. After stirring for 20 h, the mixture was poured into water and extracted using diethyl ether (3 × 150 mL). The organic layer was washed with brine, dried over Na_2_SO_4_, and filtered. The solvent was evaporated, and the residue was purified using silica gel chromatography (4:1:0.1 toluene/acetone/methanol) to give 6.89 g (80.0%) of 1 as a white powder: m.p. 170–173 °C; ^1^H-NMR (400 MHz, DMSO-*d*_6_) δ 9.30 (1H, s, OH), 9.11 (1H, OH), 9.06 (1H, OH), 8.86 (1H, s, OH), 6.93 (1H, s, aromatic-H), 6.50 (1H, s, aromatic-H), 5.94 (1H, d, *J* = 2.0 Hz, aromatic-H), 5.78 (1H, d, *J* = 2.0 Hz, aromatic-H), 4.38 (1H, d, *J* = 9.2 Hz, 2-H), 3.95 (2H, q, *J* = 7.2 Hz, CH_2_CH_3_), 3.72 (1H, m, 3-H), 2.78 (1H, dd, *J* = 5.6 and 15.2 Hz, 4a-H), 2.28 (1H, dd, *J* = 10.4 and 15.2 Hz, 4b-H), 2.01 (2H, t, *J* = 7.6 Hz, CH_2_CH_2_C=O), 1.83 (2H, t, *J* = 7.6 Hz, CH_2_CH_2_C=O), 1.45 (3H, s, Me), 1.12 (3H, t, *J* = 7.2 Hz, CH_2_C*H*_3_); ^13^C NMR (100 MHz, DMSO-*d*_6_) δ 173.0, 156.7, 156.8, 155.2, 145.1, 144.4, 131.3, 124.7, 111.9, 111.4, 99.1, 95.6, 94.2, 76.1, 72.3, 65.9, 59.7, 28.7, 27.5, 27.2, 26.6, 14.1; HRMS (ESI+) *m/z* [M^+^] calcd. for C_22_H_24_O_8_, 416.1471, found 416.1475.

#### 2.2.2. Ethyl 3-((6a*S*,12a*R*)-2,3,8,10-tetrakis((*tert*-butyldimethylsilyl)oxy)-5-methyl-5,6a,7,12a-tetrahydroisochromeno[4,3-*b*]chromen-5-yl)propanoate (**2**)

To a solution of 1 (5.18 g, 12.4 mmol) in *N*,*N*-dimethylformamide (100 mL), *tert*-butyldimethylchlorosilane (11.25 g, 74.6 mmol) and imidazole (5.08 g, 74.6 mmol) were added at 0 °C under argon. After stirring at room temperature for 20 h, the mixture was diluted with ethyl acetate, washed with brine, and dried over sodium sulfate (Na_2_SO_4_). The solvent was evaporated, and the residue was subjected to silica gel column chromatography (1:3, *n*-hexane/ethyl acetate) to give 8.44 g (78% yield) of 2 as a white solid: m.p. 104–106 °C; ^1^H-NMR (400 MHz, CHCl_3_) δ7.07 (1H, s, aromatic-H), 6.57 (1H, s, aromatic-H), 6.42 (1H, d, *J* = 2.0 Hz, aromatic-H), 6.29 (1H, d, *J* = 2.0 Hz, aromatic-H), 4.55 (1H, d, *J* = 8.8 Hz, 2-H), 4.04 (2H, q, *J* = 7.2 Hz, CH_2_CH_3_), 3.88 (1H, m, 3-H), 3.13 (1H, m, 4a-H), 2.59 (1H, m, 4b-H), 2.11 (2H, m, CH_2_CH_2_C=O), 1.82 (2H, m, C*H*_2_CH_2_C=O), 1.49 (3H, s, Me), 1.18 (3H, t, *J* = 7.2 Hz, CH_2_CH_3_), 1.02 (18H, s, 2*t*Bu), 0.99 (9H, s, *t*Bu), 0.98 (9H, s, *t*Bu), 0.24 (12H, s, 2diMe), 0.19 (6H, s, diMe), 0.180 (6H, s, diMe); ^13^C NMR (100 MHz, CDCl_3_) δ 177.1, 157.3, 155.4, 154.4, 146.4, 145.3, 131.6, 127.5, 120.2, 119.7, 108.7, 103.8, 97.5, 85.2, 80.6, 73.5, 62.6, 37.8, 32.9, 31.3, 31.1, 26.7, 26.5, 25.1, 14.1, −0.8, −1.1; HRMS (ESI+) *m/z* [M^+^] calcd. for C_46_H_80_O_8_Si_4_, 872.4930, found 872.4938.

#### 2.2.3. Ethyl 3-((6a*S*,12a*R*)-2,3,8,10-tetrakis((*tert*-butyldimethylsilyl)oxy)-5-methyl-5,6a,7,12a-Tetrahydroisochromeno[4,3-*b*]Chromen-5-yl)Propan-1-ol (**3**)

A solution of 2 (6.81 g, 7.8 mmol) in dry THF (100 mL) was added dropwise to a solution of 1M lithium aluminum hydride (LiAlH_4_) in THF (11.7 mL, 11.7 mmol) and dry THF (20 mL) at 0 °C and then stirred for 20 h at room temperature. The mixture was cooled on ice, and saturated potassium sodium tartrate was added dropwise; subsequently, the mixture was filtered through a small pad of Celite and thoroughly washed with ethyl acetate. The organic solution was washed with brine, dried over Na_2_SO_4_, and filtered. The solvent was evaporated, and the residue was purified using silica gel chromatography (1:6 *n*-hexane/ethyl acetate) to give 3.24 g (50%) of **3** as a white powder: m.p. 144–147 °C; ^1^H-NMR (400 MHz, CHCl_3_) δ 7.10 (1H, s, aromatic-H), 6.48 (1H, s, aromatic-H), 6.09 (1H, d, *J* = 2.2 Hz, aromatic-H), 5.97 (1H, d, *J* = 2.2 Hz, aromatic-H), 4.51 (1H, d, *J* = 9.6 Hz, 2-H), 3.87 (1H, m, 3-H), 3.55 (2H, m, CH_2_CH_2_CH_2_OH), 3.02 (1H, dd, *J* = 6.0 and 15.6Hz, 4a-H), 2.51 (1H, dd, *J* = 10.8 and 15.6 Hz, 4b-H), 1.91 (2H, m, CH_2_CH_2_CH_2_OH), 1.53 (3H, s, Me), 1.28 (2H, m, CH_2_CH_2_CH_2_OH), 1.01 (18H, s, 2*t*Bu), 0.99 (9H, s, *t*Bu), 0.98 (9H, s, *t*Bu), 0.29 (12H, s, 2diMe), 0.19 (6H, s, diMe), 0.18 (6H, s, diMe); ^13^C NMR (100 MHz, CDCl_3_) δ 157.8, 153.3, 152.0, 141.6, 140.4, 129.4, 126.9, 118.5, 115.1, 107.3, 104.6, 96.1, 85.0, 84.2, 76.1, 60.5, 39.6, 33.3, 31.3, 31.1, 27.3, 26.7, 26.5, 24.4, −0.8, −1.1; HRMS (ESI+) *m/z* [M + H^+^] calcd. for C_44_H_79_O_7_Si_4_, 831.4904, found 831.4910.

#### 2.2.4. *tert*-butyl (*tert*-butoxycarbonyl)(3-((6a*S*,12a*R*)-2,3,8,10-tetrakis((*tert*-butyldimethylsilyl)Oxy)-5-methyl-5,6a,7,12a-tetrahydroisochromeno[4,3-*b*]chromen-5-yl)propyl)carbamate (**4**)

To a stirred solution of 3 (5.99 g, 7.2 mmol), di-tert-butyl iminodicarboxylate (6.26 g, 28.8 mmol), and triphenylphosphine (7.57 g, 28.8 mmol) in 100 mL of dry THF, 40% diethyl azodicarboxylate in toluene (13.11 mL, 28.8 mmol) was slowly added under argon at 0 °C. After stirring at room temperature for 20 h, the mixture was diluted with ethyl acetate, washed with brine, and dried over Na_2_SO_4_. The solvent was evaporated, and the residue was subjected to silica gel column chromatography (1:10, *n*-hexane/ethyl acetate) to give 5.57 g (75% yield) of **4** as a white solid: m.p. 67–72 °C; ^1^H NMR (400 MHz, CDCl_3_) δ 7.08 (1H, s, aromatic-H), 6.46 (1H, s, aromatic-H), 6.10 (1H, d, *J* = 2.2 Hz, aromatic-H), 5.97 (1H, d, *J* = 2.2 Hz, aromatic-H), 4.43 (1H, d, *J* = 9.2 Hz, 2-H), 3.80 (1H, m, 3-H), 3.49 (2H, m, CH_2_CH_2_CH_2_NH), 3.01 (1H, dd, *J* = 6.0 and 15.8 Hz, 4a-H), 2.47 (1H, dd, *J* = 10.4 and 15.8 Hz, 4b-H), 1.75 (2H, m, CH_2_CH_2_CH_2_NH), 1.53 (2H, m, CH_2_CH_2_CH_2_NH), 1.51 (3H, s, Me), 1.46 (9H, s, *t*Bu), 1.01 (18H, s, 2*t*Bu), 0.98 (18H, s, 2*t*Bu), 0.25 (12H, s, 2diMe), 0.18 (12H, s, 2diMe); ^13^C NMR (100 MHz, CDCl_3_) δ 159.3, 156.6, 154.9, 150.5, 144.4, 140.3, 131.8, 127.6, 117.3, 115.6, 108.7, 106.1, 152.3, 83.9, 82.8, 80.5, 73.4, 44.6, 37.5, 33.0, 31.0, 30.9, 28.1, 27.3, 26.7, 26.5, 17.5, −0.8, −1.1; HRMS (ESI+) *m/z* [M + H^+^] calcd. for C_54_H_96_NO_10_Si_4_, 1030.6112, found 1030.6121.

#### 2.2.5. *N*-(3-((6a*S*,12a*R*)-2,3-bis((tert-butyldimethylsilyl)oxy)-8,10-dihydroxy-5-methyl-5,6a,7,12a-tetrahydroisochromeno[4,3-*b*]chromen-5-yl)propyl)-6-hydroxy-2,5,7,8-tetramethylchromane-2-carboxamide (**5**)

To a solution of 4 (2.10 g, 2.1 mmol) in dichloromethane (200 mL), trifluoroacetic acid (20 mL) was slowly added at 0 °C under argon. After stirring at room temperature for 2.5 h, the solvent was evaporated. The residue containing the amine was dried in vacuum and then used in the next reaction. To a solution of TrOH (0.48 g, 1.9 mmol) in *N*,*N*-dimethylformamide (20 mL) was added *N*,*N*-diisopropylethylamine (1.1 mL, 6.4 mmol) followed by 1-[Bis(dimethylamino)methylene]-1H-1,2,3-triazolo[4,5-*b*]pyridinium3-Oxide Hexafluorophosphate (HATU: 1.61 g, 4.2 mmol). The mixture was stirred at room temperature for 15 min, after which the dark clear mixture was treated with the solution of the amine and *N*,*N*-diisopropylethylamine (1.1 mL, 6.4 mmol) in DMF (20 mL). After stirring at room temperature for 24 h, the mixture was diluted with ethyl acetate, washed with brine, and dried over Na_2_SO_4_, and the solvent was evaporated. The residue was purified by silica gel column chromatography (1:1, *n*-hexane/ethyl acetate) to give 0.67 g (30% yield) of **5** as a white solid: m.p. 167–170 °C; ^1^H NMR (400 MHz, DMSO-*d*_6_) δ 9.35 (1H, s, OH), 9.05 (1H, s, OH), 7.47 (1H, s, OH), 7.11 (1H, m, NH), 7.01 (1H, s, aromatic-H), 6.51 (1H, s, aromatic-H), 5.95 (1H, d, *J* = 2.2 Hz, aromatic-H), 5.80 (1H, d, *J* = 2.2 Hz, aromatic-H), 4.41 (1H, d, *J* = 8.4 Hz, 2-H), 3.74 (1H, m, 3-H), 2.98 (2H, m, CH_2_CH_2_CH_2_NH), 2.83 (1H, m, 4a-H), 2.42 (2H, m, CH_2_CH_2_), 2.30 (1H, m, 4a-H), 2.11 (2H, m, CH_2_CH_2_), 2.10 (3H, s, Me), 2.08 (3H, s, Me), 2.00 (3H, s, Me), 1.71 (2H, m, CH_2_CH_2_CH_2_NH), 1.55 (2H, m, CH_2_CH_2_CH_2_NH), 1.35 (3H, s, Me), 0.98 (9H, s, *t*Bu), 0.94 (9H, s, *t*Bu), 0.21 (6H, s, diMe), 0.18 (6H, s, diM); ^13^C NMR (100 MHz, CDCl_3_) δ 177.8, 157.7, 156.8, 156.0, 147.3, 146.8, 146.0, 145.7, 132.7, 129.3, 128.0, 121.5, 118.5, 117.7, 116.3, 114.0, 97.4, 96.3, 94.2, 92.4, 81.9, 80.8, 76.5, 42.5, 41.0, 36.2, 33.3, 31.1, 30.0, 28.3, 26.9, 22.3, 21.0, 16.8, 12.5, 11.8, −0.8; HRMS (ESI+) *m/z* [M + H^+^] calcd. for C_46_H_68_NO_9_Si_2_, 834.4433, found 834.4439.

#### 2.2.6. 6-hydroxy-2,5,7,8-tetramethyl-*N*-(3-((6a*S*,12a*R*)-2,3,8,10-tetrahydroxy-5-methyl-5,6a,7,12a-tetrahydroisochromeno[4,3-*b*]chromen-5-yl)propyl)chromane-2-carboxamide (PCat-TrOH)

To a solution of **5** (0.76 g, 0.72 mmol) in dry THF (200 mL), 1M tetrabutylammonium fluoride in THF (3.93 mL, 3.93 mmol) was slowly added at 0 °C under argon. After stirring at room temperature for 1.5 h, the mixture was diluted with ethyl acetate, washed with brine, and dried over Na_2_SO_4_. The solvent was evaporated, and the residue was purified by column chromatography (2:1:0.1, toluene/acetone/methanol to give 0.22 g (50% yield) of PCat-TrOH as a white solid: m.p. 186–189 °C; ^1^H NMR (400 MHz, DMSO-*d*_6_) δ 9.29(1H, s, OH), 9.05 (2H, s, OH), 8.74(1H, s, OH), 7.48 (1H, s, OH), 7.16 (1H, m, NH), 6.90 (1H, s, aromatic-H), 6.38 (1H, s, aromatic-H), 5.94 (1H, s, aromatic-H), 5.78 (1H, s, aromatic-H), 4.34 (1H, m, 2-H), 3.69 (1H, m, 3-H), 2.95 (2H, m, CH_2_CH_2_CH_2_NH)), 2.80 (1H, m, 4a-H), 2.42 (H, m, CH_2_CH_2_), 2.31 (1H, dd, *J* = 10.4 and 14.8 Hz, 4b-H), 2.14 (2H, m, CH_2_CH_2_), 2.09 (3H, s, Me), 2.07 (3H, s, Me), 1.99 (3H, s, Me), 1.68 (2H, m, CH_2_CH_2_CH_2_NH), 1.51 (2H, m, CH_2_CH_2_CH_2_NH), 1.34 (3H, s, Me); ^13^C NMR (100 MHz, CDCl_3_) δ 177.6, 157.8, 156.6, 155.6, 146.5, 148.8, 145.0, 144.0, 132.7, 129.0, 127.5, 121.0, 117.5, 116.0, 114.1, 113.0, 98.0, 96.8, 94.8, 92.4, 81.0, 79.2, 74.5, 42.0, 40.2, 36.0, 34.4, 32.4, 26.9, 22.3, 21.2, 16.8, 12.6, 11.5; HRMS (ESI+) *m/z* [M + H^+^] calcd. for C_34_H_40_NO_9_, 606.2704, found 606.2709.

### 2.3. Antioxidant Activity Measurements

Galvinoxyl radical (GO^•^) was commercially obtained from Tokyo Chemical Industry Co., Ltd., (Tokyo, Japan); GO^•^ is the highest available purity and was used without further purification unless otherwise noted. Acetonitrile (MeCN), used as solvent, was commercially obtained from Nacalai Tesque Inc. (Kyoto, Japan) and used as received. The rates of GO^•^-scavenging reactions using test samples in MeCN were determined by monitoring the absorbance change at 428 nm due to GO^•^ (ε = 1.4 × 10^5^ M^−1^ cm^−1^) after mixing it in MeCN at a volumetric rate of 1:1 using a stopped-flow technique on a UNISOKU REP-1000-02 NM spectrophotometer (UNISOKU Co., Ltd., Osaka, Japan) at 298 K. The pseudo-first-order rate constants (*k*_obs_) were determined by a least-squares curve fitting using an UNITCOM SOLUTION-M06M-134-UHX (UNIT.COM Inc., Osaka, Japan). The first-order plots of ln(*A*–*A*_∞_) vs. time (*A* and *A*_∞_ were denoted as the absorbance at the reaction time and the final absorbance, respectively) were linear with the correlation coefficient *ρ* > 0.999 until three or more half-lives. In all cases, the *k*_obs_ values obtained from at least three independent measurements agreed within ±5% experimental error. In all cases, the solutions were normally equilibrated with air.

### 2.4. Spectral Titrations

A total of 5 μL of PCat-TrOH (2.63 × 10^−4^ M) in MeCN was added to GO^•^ (7.77 × 10^−6^ M) in MeCN, and the absorbance at 428 nm due to GO^•^ was plotted with respect to the concentration ratio, [PCat-TrOH]/[GO^•^]. The UV–vis spectral changes accompanying the reaction were monitored and measured using an Agilent 8453 photodiode array spectrophotometer thermostated with a Peltier temperature control at 298 K (Agilent Technologies, Santa Clara, CA, USA). In all cases, the solutions were normally equilibrated with air.

## 3. Results

### 3.1. Chemistry

We designed and synthesized a compound, PCat-TrOH, in which TrOH was conjugated to PCat, creating a molecule with multiple radical-scavenging structures (Figure 2). The PCat site, where the 3D catechin structure was fixed, was obtained through the oxa-Pictet–Spengler reaction between (+)-catechin and ethyl levulinate. Specifically, **1** was synthesized by treating a tetrahydrofuran (THF) solution of catechin and ethyl levulinate with trimethylsilyl trifluoromethanesulfonate (TMSOTf). Next, the four hydroxyl groups were protected with *tert*-butyldimethylchlorosilyl (TBS) groups, after which the ester structure was reduced with lithium aluminum hydride (LiAlH_4_) to obtain alcohol **3**. Then, **3** underwent a Mitsunobu reaction using (Boc)_2_NH as a pronucleophile in the presence of Ph_3_P and DEAD to obtain **4**. Trifluoroacetic acid (TFA) was used to remove the BOC group together with the TBS groups on A ring, after which the amino group of **4** underwent a condensation reaction with the carboxyl group of TrOH, using the condensing agent DIPEA/HATU without purification to obtain compound **5**. Finally, TBAF was used for deprotection, resulting in PCat-TrOH.

### 3.2. Radical-Scavenging Activity

The radical-scavenging activities of PCat-TrOH, TrOH, PCat, and Cat were evaluated in acetonitrile using galvinoxyl radical (GO^•^), a model compound for ROS. The strong absorption of GO^•^ at 428 nm immediately disappeared upon reaction with the antioxidant. Therefore, the change in color of the GO^•^ solution in the presence of the antioxidant was used to measure the radical-scavenging activity rates of PCat-TrOH, TrOH, PCat, and Cat. This spectral change was monitored by measuring the decrease in absorbance at 428 nm, using the stopped-flow method (Figure 3). The concentrations of PCat-TrOH, TrOH, PCat, and Cat were set to over 10 times the concentration of GO^•^ to measure the decrease in absorbance at 428 nm under pseudo-first-order reaction conditions, allowing us to obtain the pseudo-first-order rate constant (*k*_obs_). The *k*_obs_ value increased linearly with increasing concentrations of PCat-TrOH, TrOH, PCat, and Cat (Figure 4). The second-order rate constant (*k*) in Equation (1) could be determined from the slope of the plot (Equation (2)) for the GO^•^-scavenging reaction by PCat-TrOH in MeCN to be 9.6 × 10^2^ M^−1^ s^−1^. The *k* values of TrOH and PCat were also determined in a similar manner to be 1.1 × 10^3^ M^−1^ s^−1^ and 1.7 × 10^2^ M^−1^ s^−1^, respectively.
−d[GO^•^]/d*t* = *k*[PCat-TrOH][GO^•^](1)
*k*_obs_ ([PCat-TrOH] > 10[GO^•^]) = *k*[PCat-TrOH](2)

We have also previously reported that the *k* of Cat under the same conditions was 2.6 × 10^1^ M^−1^ s ^−1^. The *k* value of PCat was approximately 6.5 times that of Cat, indicating that the planar structure increased its radical-scavenging activity. Meanwhile, TrOH was even more powerful than PCat, with a *k* value approximately 6.5 times greater than that of Cat. The *k* value of the newly synthesized compound PCat-TrOH, which binds TrOH to PCat, was approximately 5.6 times larger than that of planar catechin, exhibiting an almost equally powerful radical-scavenging activity as that of TrOH alone.

### 3.3. Spectral Titrations

ROS are eliminated by one-electron reduction, so catechin eliminates two ROSs and is oxidized to quinone. Additionally, α-tocopherol reduces two lipid peroxides and is itself oxidized to tocopheryl quinone. Therefore, we sought to clarify the number of GO^•^ radicals that could be eliminated by PCat-TrOH (antioxidant capacity) through titration (Figure 5), which we conducted by adding antioxidants to a GO^•^ solution, plotting the reduced 428 nm absorbance of GO^•^ and finally obtaining the molar ratio of antioxidants to GO^•^ when GO^•^ was completely eliminated. As shown in Figure 6, in the spectral titration conducted in this study, the 428 nm absorbance of GO^•^ was completely eliminated by 0.25 times the quantity of PCat-TrOH relative to GO^•^, indicating that the molar ratio of PCat-TrOH to GO^•^ was 1:4. Meanwhile, the 428 nm absorbance was eliminated by adding 0.5 times the quantity of PCat or TrOH relative to GO^•^, indicating that their molar ratios were 1:2. Therefore, it was thought that PCat-TrOH eliminated four GO^•^ radicals, compared to PCat or TrOH, which eliminated only two GO^•^ radicals.

## 4. Discussions

Natural antioxidants exert their effects through both direct scavenging of various radical species produced in the body and by activating different antioxidant enzymes. However, the direct radical-scavenging activity against reactive oxygen species (ROS) is considered the essential antioxidant function of these compounds. In this study, we focused on the direct radical-scavenging activity of antioxidants against ROS and synthesized a compound, PCat-TrOH, which combines the potent antioxidant PCat, having stronger antioxidant activity than natural catechins, with the Trolox structure, the core of vitamin E’s antioxidant function. The PCat-TrOH had a *k* value approximately 5.6 times higher than that of PCat, indicating an antioxidant effect nearly as strong as TrOH. This result suggests that the radical-scavenging activity of PCat-TrOH may preferentially involve hydrogen transfer from TrOH. Meanwhile, the titration results showed that four GO^•^ radicals were eliminated for every PCat-TrOH molecule. This suggests that PCat-TrOH eliminates GO^•^ through the reduction effects of both TrOH and PCat. The fact that TrOH exhibits a radical-scavenging activity approximately 6.5 times stronger than that of PCat and that PCat-TrOH exhibits a similar level of antioxidant activity as TrOH, suggests that after TrOH in PCat-TrOH eliminates GO^•^ through hydrogen transfer, the catechol-derived hydrogen of PCat is transferred to the oxidized TrOH, regenerating TrOH in its reduced form. Detailed analyses are currently underway to clarify the mechanism of radical elimination by this intramolecular network. The above results show that PCat-TrOH possesses excellent antioxidant properties, with powerful radical-scavenging activity nearly equivalent to that of TrOH and a radical-scavenging capacity double that of TrOH.

Vitamin E, particularly α-tocopherol, can trap lipid radicals and lipid peroxide radicals, which play an important role in lipid peroxidation chain reactions, and terminate free radical chain reactions [23,24]. These substances capture radicals through a hydrogen transfer reaction from the hydroxyl group on the aromatic ring of α-tocopherol, oxidizing themselves to an α-tocopherol radical [25]. Extracellular ascorbic acid (vitamin C) exerts its antioxidant effect in biological membranes, reducing the tocopheroxyl radicals produced and regenerating α-tocopherol [26]. This regenerative action allows α-tocopherol to exhibit a synergistic effect as an antioxidant, breaking free radical chains and providing excellent protection against oxidative stresses in the body. Following the elimination of ROS radicals through reduction, catechin is oxidized to radical catechin and subsequently converts to a two-electron oxidized quinone [27]. Ascorbic acid (vitamin C) also exhibits a reducing effect on these oxidized products, potentially regenerating catechin. Therefore, PCat-TrOH is thought to be regenerated by ascorbic acid reducing the radicals generated by the oxidation of PCat and TrOH during radical-scavenging activity. Phenolic compounds with strong antioxidant activity typically possess high electron density in their phenolic moiety, making the quinone forms, produced via antioxidant reactions, difficult to reduce back to their phenol structures. In the case of PCat-TrOH, although TrOH exhibits stronger antioxidant activity compared to PCat, the oxidized form of TrOH generated through antioxidant reactions is less susceptible to reduction by endogenous reductants compared to the oxidized form of PCat. Consequently, when both the PCat and TrOH moieties of PCat-TrOH are oxidized, the PCat moiety is preferentially reduced by biological reductants. It is expected that the regenerated PCat can subsequently reduce the oxidized TrOH through an intramolecular electron transfer network. As a result of this structural interplay, PCat-TrOH demonstrates antioxidant activity comparable to that of TrOH, with double the antioxidant capacity. Furthermore, the oxidized form of PCat-TrOH, produced during antioxidant reactions, shows superior regeneration by biological reductants such as ascorbic acid compared to TrOH. PCat-TrOH, which exhibits high antioxidant activity and capacity, is expected to provide an extremely high protective effect through the regenerative effect of extracellular ascorbic acid on lipid peroxidation chain reactions in biological membranes. The analysis of the antioxidant effect of PCat-TrOH against oxidative stress in cells and its mechanism is currently under investigation.

## 5. Conclusions

In the present study, we successfully synthesized PCat-TrOH, in which TrOH, known for its antioxidant properties, is conjugated to PCat. While TrOH exhibited approximately 6.5 times greater radical-scavenging activity than PCat, the conjugate PCat-TrOH, which combines PCat with TrOH, also demonstrated similarly potent radical-scavenging activity to TrOH. Furthermore, PCat-TrOH exhibited double the radical-scavenging capacity against reactive oxygen species (ROS) compared to either PCat or TrOH. These results suggest that catechin, when stabilized in a planar conformation, shows enhanced radical-scavenging activity and that conjugation with TrOH further enhances both its antioxidant activity and capacity. PCat-TrOH may exhibit strong antioxidant effects through the formation of an intramolecular antioxidant network, involving hydrogen transfer reactions between the TrOH and PCat moieties. Detailed analyses to clarify this possibility, as well as biological tests on the antioxidant activity of PCat-TrOH using cells, are currently underway. Furthermore, the results of this study suggest that the addition of antioxidant compounds to planar catechin may further enhance antioxidant activity and promote the formation of intramolecular antioxidant networks. Consequently, we plan to synthesize compounds containing multiple antioxidants found in the body, such as lipoic acid, in future studies.

## Figures and Tables

**Figure 1 antioxidants-13-01165-f001:**
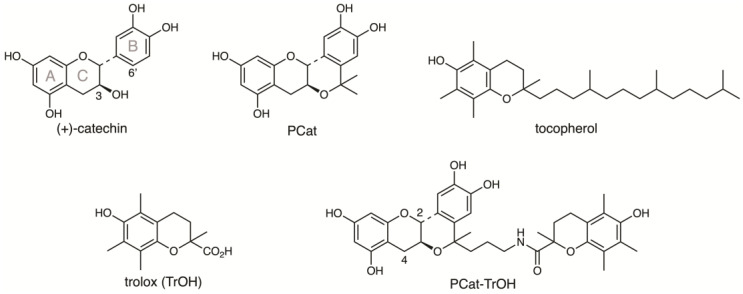
Structures of (+)-catechin, PCat, α-tocopherol, TrOH, and PCat-TrOH.

**Figure 2 antioxidants-13-01165-f002:**
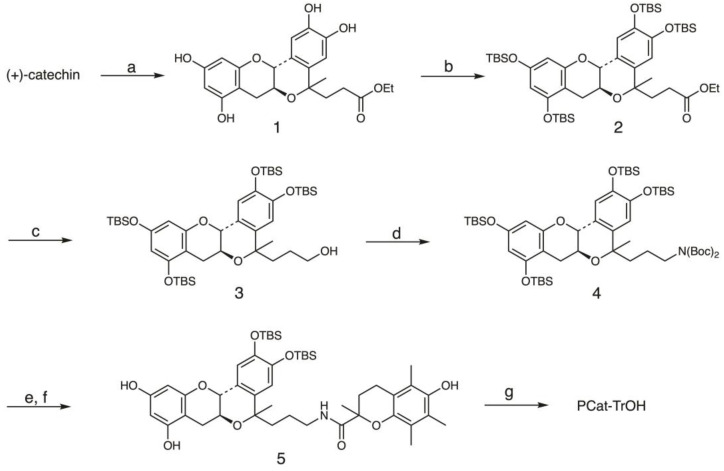
Synthesis of PCat-TrOH. Regents and conditions: (**a**) ethyl levulinate, TMSOTf, THF, −30 °C; (**b**) *tert*-butyldimethylchlorosilane, imidazole, DMF, 0 °C → rt; (**c**) LiAlH_4_, THF, 0 °C → rt; (**d**) di-*tert*-butyliminodicarboxylate, Ph_3_P, DEAD,THF, 0 °C → rt; (**e**) TFA, CH_2_Cl_2_, 0 °C → rt; (**f**) TrOH, DIPEA, HATU, DMF, 0 °C → rt; (**g**) TBAF, CH_2_Cl_2_, 0 °C → rt.

**Figure 3 antioxidants-13-01165-f003:**
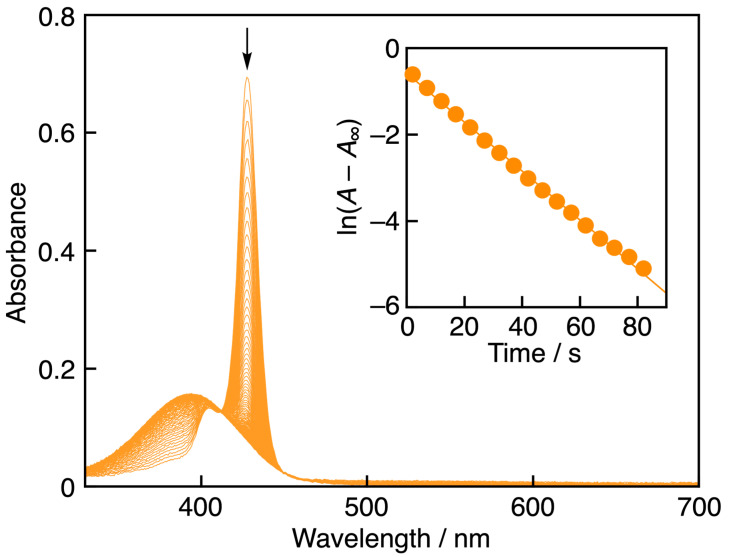
Spectral change (interval: 1 s) observed during the reaction of PCat-TrOH (7.2 × 10^−5^ M) with GO^•^ (5.6 × 10^−6^ M) in MeCN at 298 K. The arrow denotes the direction of absorbance changes. Inset: The first-order plot of the absorbance at 428 nm.

**Figure 4 antioxidants-13-01165-f004:**
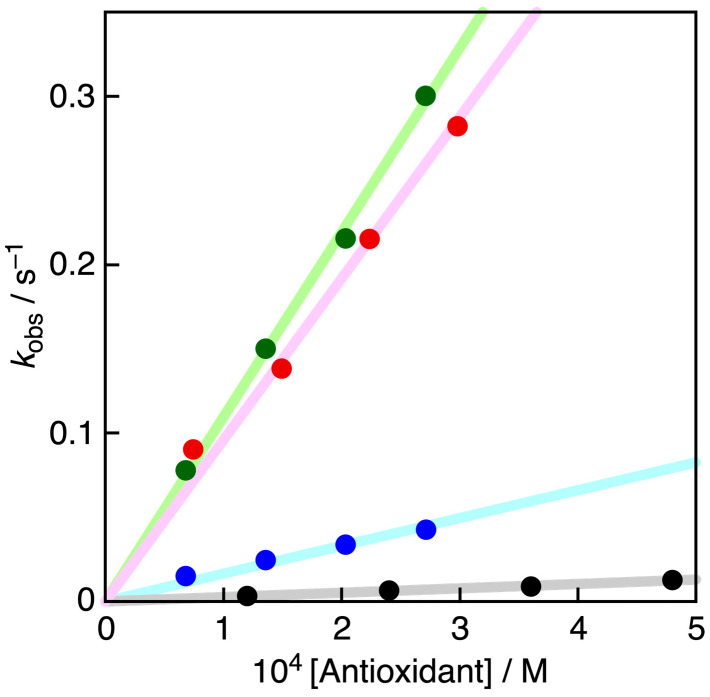
Plots of pseudo-first-order rate constants (*k*_obs_) vs. concentrations of TrOH (green circles), PCat-TrOH (red circles), PCat (blue circles), and Cat (black circles) in MeCN at 298 K.

**Figure 5 antioxidants-13-01165-f005:**
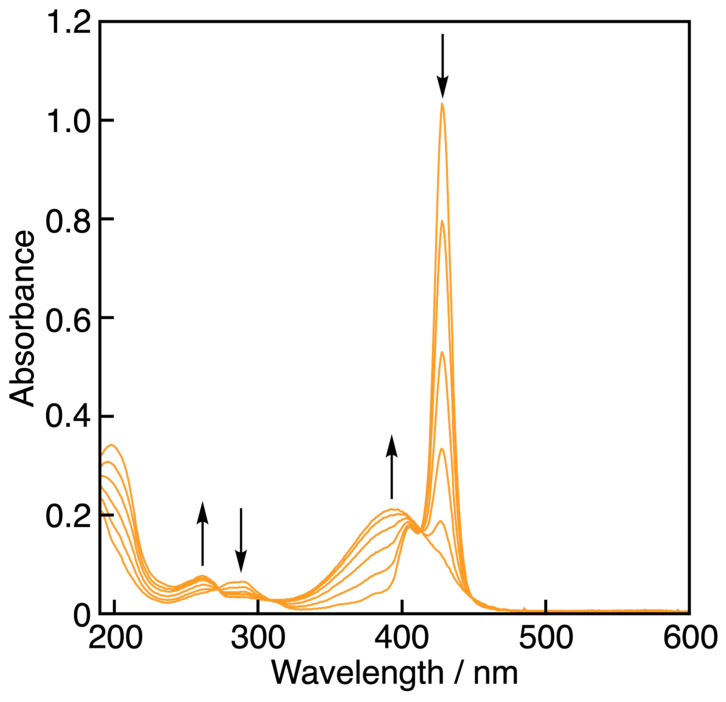
Spectral change observed upon addition of PCat-TrOH (0, 4.4 × 10^−7^, 8.7 × 10^−7^, 1.3 × 10^−6^, 1.7 × 10^−6^, 2.2 × 10^−6^ M) to GO^•^ (7.8 × 10^−6^ M) in MeCN at 298 K. The arrows denote the direction of absorbance changes.

**Figure 6 antioxidants-13-01165-f006:**
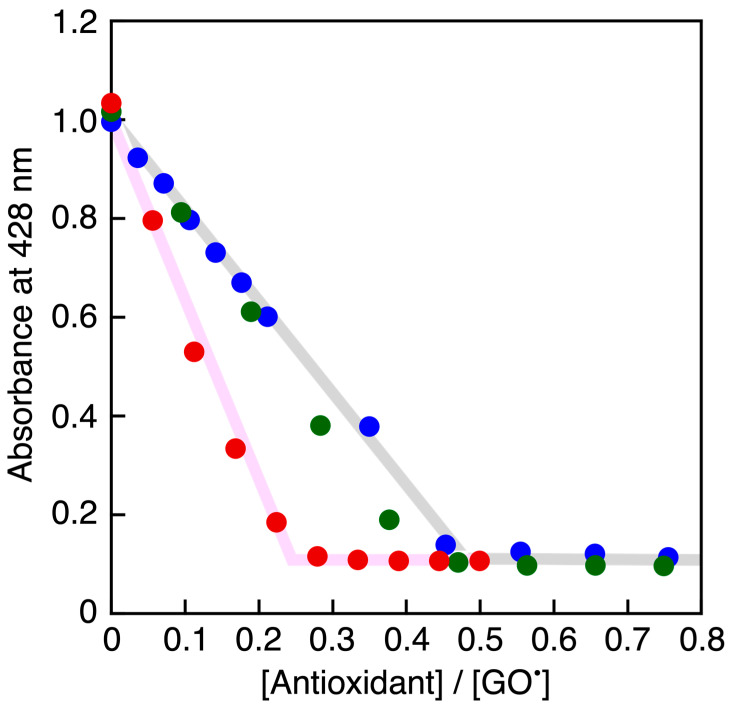
Plots of the absorbance at 428 nm vs. the concentration ratios of antioxidants (TrOH (green circles), PCat-TrOH (red circles), and PCat (blue circles)) to GO^•^, [antioxidants]/[GO^•^], in MeCN.

## Data Availability

All data are contained in this article.

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
