# Peer review of "Antioxidant Activity of Planar Catechin Conjugated with Trolox"

_antioxidants, 2024, doi:10.3390/antiox13101165_

Round 1

Reviewer 1 Report

The authors report synthesis of a compound PCat–TrOH by binding Trolox (TrOH), an α-tocopherol analog, to the planar catechin (PCat), a natural antioxidant with a fixed 3D catechin structure on a plane. Radical-scavenging activity of PCat is five times higher than that of catechin and radical-scavenging activity of TrOH is 6.5 times higher than that of PCat, the combination of these two moieties into one molecule doubled the radical-scavenging activity against reactive oxygen species compared to PCat or TrOH. Authors suggest that the new compound will exhibit an excellent antioxidant effect against lipid peroxidation. The manuscript represents a constructive approach to development of a new molecule based on the known antioxidant properties of the contributing initial compounds. My suggestion is to accept the manuscript for publication after the following minor revisions:

-      -  My suggestion is to add information about the attribution of the proton signals to the protons in the molecule in the 1H NMR spectra in the Experimental part.

-         - Lines 363-365: the sentence here needs reference.

-       - The reference list should be revised because incomplete information is provided for quite many references. The page numbers or article numbers are missing.

    - The text needs formatting to meet the usual style used in organic synthesis articles: numbers of all compounds should be bald, the symbols of chemical elements in the names of organic compounds are written in italics (for example, N,N-dimethylformamide), n in n-hexane should be in italics as well,

-      -  Line 229: alcohol (3) should be written as alcohol 3 because compound number is written in brackets only when it follows the full name of the compound.

-       - Line 235: the code name for the compound should be corrected. Now it is written as PCAT–TrOH..

-      -  Line 318: my suggestion is to write two ROSs (plural).

Author Response

Comments1: My suggestion is to add information about the attribution of the proton signals to the protons in the molecule in the 1H NMR spectra in the Experimental part.

Response 1: In accordance with the reviewer’s comments, the assignments of the proton signals to the proton in the molecule in the 1H NMR spectra were added to the Experimental section.

Comments 2: Lines 363-365: the sentence here needs reference.

Response 2: Lines 363–365 were removed when revising the Discussion. However, the content has already been described in the Introduction, along with the appropriate references.

Comments 3: The text needs formatting to meet the usual style used in organic synthesis articles: numbers of all compounds should be bald, the symbols of chemical elements in the names of organic compounds are written in italics (for example, N,N-dimethylformamide), n in n-hexane should be in italics as well.

Response 3: In accordance with the reviewer’s comments, the manuscript has been revised to align with the general style of organic synthetic articles.

Additionally, all other points raised by the reviewer have been addressed.

Reviewer 2 Report

The present work represents an important contribution to the field of antioxidants, but this article only presents the main features of the research. The authors have used only one method to determine antioxidant activity, and the discussion of the detailed mutual dependence of molecular structure and mechanism of action for all four compounds included in the study is missing. The first 10 lines of the discussion section are a repetition of the results and half of the discussion is based on predictions. I think it is necessary to expand the results and discussion with some insights into what is "still under investigation".

Fig 3 is not self-explanatory; please explain what A and A∞ refer to, or add an equation for calculating the change in absorbance in the Materials and Methods section.
Some conclusions are supported by results, while others are based purely on speculation (at least what can be seen from results obtained in present study).

Author Response

Major comments: The present work represents an important contribution to the field of antioxidants, but this article only presents the main features of the research. The authors have used only one method to determine antioxidant activity, and the discussion of the detailed mutual dependence of molecular structure and mechanism of action for all four compounds included in the study is missing. The first 10 lines of the discussion section are a repetition of the results and half of the discussion is based on predictions. I think it is necessary to expand the results and discussion with some insights into what is "still under investigation".

Detail comments: Some conclusions are supported by results, while others are based purely on speculation (at least what can be seen from results obtained in present study).

Response: In general, natural antioxidants not only exert direct scavenging effects on reactive oxygen species (ROS) but also activate various antioxidant enzymes in living organisms. Moreover, many natural antioxidants exhibit diverse biological activities beyond antioxidant activity, such as regulation of lipid metabolism, antibacterial, and antiviral effects. In this study, we focused on the direct radical scavenging effect of antioxidants on ROS. We synthesized novel compounds containing multiple antioxidant moieties within a single molecule to achieve enhanced antioxidant activity and capacity through an intramolecular electron-transfer network. Kinetic analysis of radical scavenging activity, using model ROS compounds, is the only method to elucidate the physicochemical radical scavenging capacity of antioxidants. Using this approach, we investigated the radical scavenging activity and capacity of four compounds, specifically characterizing the antioxidant properties of PCat-TrOH. At present, we are also examining the cellular antioxidant activity of these four compounds, as well as their glucosidase inhibitory activity and their antiproliferative effects against cancer cells. We plan to report on the antioxidant characteristics of PCat-TrOH based on these findings in the near future.

In response to the reviewers' comments, we revised the paper by adding a statement at the beginning of the Discussion section clarifying that the method used to evaluate antioxidant activity in this study was the approach that could elucidate radical scavenging activity and radical scavenging capacity through kinetic analysis. In addition, we added a discussion on the superiority of PCat-TrOH based on the detailed mutual dependence between molecular structure and redox mechanism in the regeneration of the original antioxidants by the reducing substances after the four compounds were oxidized by radical scavenging. The conclusion has also been revised to clearly articulate the key findings that can be understood from the results of this study and the insight into what is ''still under investigation''

Detail comments: Fig 3 is not self-explanatory; please explain what A and A∞ refer to, or add an equation for calculating the change in absorbance in the Materials and Methods section.

Response: In the results section, a description of how to determine the reaction rate of radical scavenging activity based on changes in absorption was added, citing the relevant reaction equations.

Round 2

Reviewer 2 Report

Dear Authors,

Your revision version has been improved, so I have no further comments.

Dear Authors,

Your revision version has been improved, so I have no further comments.

Wish you all the best,

Reviewer